# A model explaining mRNA level fluctuations based on activity demands and RNA age

Zhongneng Xu[1,2]*, Shuichi Asakawa[2]

**1** Department of Ecology, Jinan University, Guangzhou, China, **2** Department of Aquatic Bioscience, Graduate School of Agricultural and Life Sciences, The University of Tokyo, Tokyo, Japan

* txuzn@jnu.edu.cn, xuzhongneng@g.ecc.u-tokyo.ac.jp

**Data Availability Statement:** All relevant data are within the manuscript and its Supporting Information files.

**Funding:** The authors received no specific funding for this work.

## Abstract

Cellular RNA levels typically fluctuate and are influenced by different transcription rates and RNA degradation rates. However, the understanding of the fundamental relationships between RNA abundance, environmental stimuli, RNA activities, and RNA age distributions is incomplete. Furthermore, the rates of RNA degradation and transcription are difficult to measure in transcriptomic experiments in living organisms, especially in studies involving humans. A model based on activity demands and RNA age was developed to explore the mechanisms of RNA level fluctuations. Using single-cell time-series gene expression experimental data, we assessed the transcription rates, RNA degradation rates, RNA life spans, RNA demand, accumulated transcription levels, and accumulated RNA degradation levels. This model could also predict RNA levels under simulation backgrounds, such as stimuli that induce regular oscillations in RNA abundance, stable RNA levels over time that result from long-term shortage of total RNA activity or from uncontrollable transcription, and relationships between RNA/protein levels and metabolic rates. This information contributes to existing knowledge.

## Author summary

Detected cellular RNA levels usually fluctuate. The understanding of the fundamental relationships between RNA level fluctuations, the rates of RNA degradation and transcription, environmental stimuli, RNA activities, and RNA age distributions is incomplete. In the present research, we developed a model based on the demands of RNA (related to intrinsic and/or extrinsic information), RNA age (determines the survival time and biological activity of an RNA), transcription, and RNA degradation to explain the mechanism underlying intracellular RNA level fluctuations. We also explored applicability of the model for analysing dynamic processes between interacting biomolecules, such as the relationship between RNA and protein level fluctuations. Using single-cell time-series gene expression experimental data, we assessed some biological parameters, such as transcription rates, RNA degradation rates, and RNA life spans. This model could also predict RNA levels under simulation backgrounds, such as stimuli that induce regular oscillations in RNA abundance, stable RNA levels over time that result from long-term shortage of total RNA activity or from uncontrollable transcription, and relationships between RNA/

**Competing interests:** The authors have declared that no competing interests exist.

protein levels and metabolic rates. This information contributes to existing knowledge and provides a new perspective for future studies.

## Introduction

Mechanisms underlying the maintenance of specific RNA levels in a given cell are a target of transcriptomics studies [1–3]. Detected cellular RNA levels usually fluctuate [4–6]. Studies have provided diverse explanations for the mechanisms of RNA level fluctuations, such as stochastic pulsing of transcription and environmental determinants [5,7–14]. However, the limitations of transcriptomic techniques mean that some potential fundamental drivers of RNA level fluctuations remain unknown.

Little is known about the relationship between the supply and demand of RNA. Excess or deficiency in RNA can be harmful [15–18]. Regulation of appropriate RNA levels to meet cellular demands for RNA appears to be necessary to maintain cellular health. Previous studies have shown that some transcription initiation events result from intrinsically and/or extrinsically deterministic factors, such as extrinsic hormones, the cell cycle, and biological rhythm signals [19–24]. The challenge involves linking RNA level fluctuations with the effects of various environmental stimuli, the demands for RNA, and transcription.

RNAs are queued along the DNA template for production by RNA polymerase. The term "RNA age", which specifies how long an RNA has existed since its initial transcription, was previously used in our study in a gene expression model [25]; later, Rodriques et al used the term in an RNA timestamp analysis [26]. From the birth to the death, RNAs are involved in temporal and spatial biological processes and in modulating various physiological activities [27–29]. Moreover, the lifespan of RNAs usually ranges from a few minutes to more than two days [30] and may be longer than the interval times between two consecutive transcriptional pulses, which usually range from a few minutes to a few hours [8,31–33]. Thus, transcripts with different RNA ages coexist in the same cellular pool of RNAs. Therefore, relative abundance and expected degradation times of RNAs at different ages should be considered in the studies of RNA activities and fluctuations in RNA abundance.

Transcription and RNA degradation rates are key parameters that lead to fluctuations in RNA levels [13,30]. Cellular RNA levels detected in an experiment are not RNA levels that are transcribed during the experimental period but rather, represent RNA abundance, which is RNA accumulation plus RNA transcription minus RNA degradation [34]. Special methods to examine transcription and RNA degradation rates have been reported [35,36], but they are difficult to use in routine RNA experiments in living organisms, especially in studies involving humans. Thus, a generally applicable method for quantitative estimation of transcriptional and RNA degradation rates is still lacking.

In the present research, we developed a model based on the demands of RNA (related to intrinsic and/or extrinsic information), RNA age (determines the survival time and biological activity of an RNA), pulse transcription, and RNA degradation to explain the mechanism underlying intracellular RNA level fluctuations. We also explored applicability of the model for analysing dynamic processes between interacting biomolecules, such as the relationship between RNA and protein level fluctuations. The aim of this study was to provide explanations for some transcriptomic phenomena and to provide a new perspective for future studies.

## Results

### Explaining mRNA level fluctuations in single-cell experimental data by the present model

This model was able to simulate experimental data of mRNA level fluctuations (including regular, partially regular, and irregular fluctuations) in single cells reported in the literature (S1 Table). The simulation results of the expression of three gene time series (Fig 1) indicated that the calculated values of $R^2$ and median absolute percentage error (MdAPE) of the model were a good fit for experimental values, thus, this model was reasonable to use to explore biological parameters contained in these experimental data. Transcription rates, RNA degradation rates, RNA demands, RNA life spans, accumulated transcription levels, and accumulated RNA degradation levels were calculated. Analysis of fluctuations in the levels of *Saccharomyces cerevisiae* HSP26 mRNA [21] using our model indicated that accumulated transcription level during the experiment (70 minutes) was 20.4 units, accumulated RNA degradation level during the experiment was 18.1 units, life span of mRNA of the HSP26 gene was 20 minutes, pulse transcription level at each time was 1.4 units, the rate of survival at RNA age 1 was 0.9, the survival rate of RNA age 2 was 0.9, and the survival rate of RNA age 3 was 0.5 (Fig 1A). Analysis of fluctuations in the level of *S. cerevisiae* YNR014W mRNA [21] indicated that accumulated transcription level during the experiment (70 minutes) was 18.9 units, accumulated RNA degradation level during the experiment was 18.9 units, life span of mRNA of the YNR014W gene was 20 minutes, pulse transcription level at each time was 3 units, the rate of survival at RNA age 1 was 0.2, the survival rate of RNA age 2 was 0.2, and the survival rate of RNA age 3 was 0 (Fig 1B). The results of analysis of fluctuations in the levels of GnRH mRNA in mouse

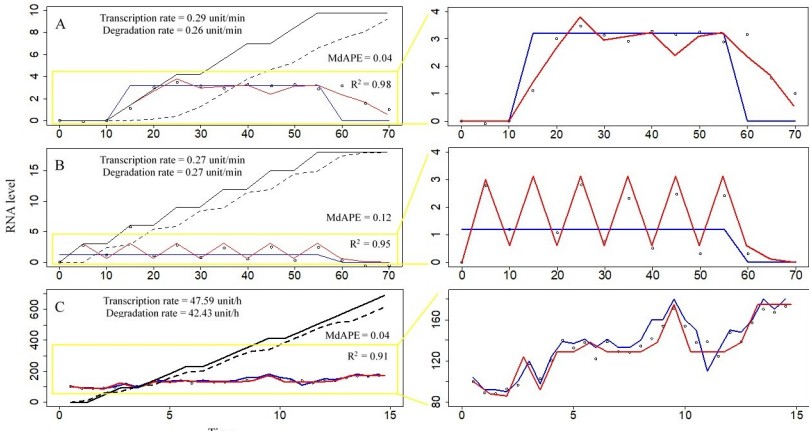

**Fig 1. Analysis of experimental data of single-cell RNA abundance using the present model.** Hollow circles represent RNA levels of experimental data from the references [21,37]. Blue line represents the demand for RNA activity (DRA). Red curve represents the RNA level of the simulation result of the model. Black solid line represents the cumulative transcription RNA level calculated by the model. Black dashed line represents the accumulated degradation RNA level calculated by the model. (A) Partially regular fluctuations: analysis of mRNA level fluctuations of the *Saccharomyces cerevisiae* HSP26 gene by the model. The experimental data have been reported by Hao and O'Shea (2011) [21]. The x-axis is time, and the unit of time is minute. The y-axis is the RNA level, and the unit is normalized fold change of mRNA level with the baseline subtracted. (B) Regular fluctuations: analysis of mRNA level fluctuations of the *S. cerevisiae* YNR014W gene by the model. The experimental data have been reported by Hao and O'Shea (2011) [21]. The x-axis is time, and the unit of time is minute. The y-axis is the RNA level, and the unit is normalized fold change of mRNA level with the baseline subtracted. (C) Irregular fluctuations: analysis of mRNA level fluctuations of the mouse GT1-1 cell GnRH gene by the model. The experimental data have been reported by Nuñez et al. (1998) [37]. The x-axis is time, and the unit of time is hour. The y-axis is the RNA level, and the unit of mRNA level is normalized photonic emissions.

GT1-1 cells [37] indicated that accumulated transcription level during the experiment (14 hours) was 690 units, accumulated RNA degradation level during the experiment was 615.2 units, life span of mRNA of the GnRH gene was 3 hours, pulse transcription level at each time was 46 units, the rate of survival at RNA age 1 was 1, the survival rate of RNA age 2 was 1, and the survival rate of RNA age 3 was 0.8 (Fig 1C). In some cases, the estimated values of the parameters of RNA metabolism calculated by the present model provided sufficient information for biological and medical purposes. Other examples of simulations of experimental data of fluctuations in mRNA levels obtained using the present model are shown in -S1 Fig.

Degradation coefficients were closely related to fluctuations in RNA levels. Under the condition of stable demands for RNA activities (related to environmental stimuli), low rates of RNA survival at various RNA ages resulted in high-frequency fluctuations in RNA levels (Fig 1B). In contrast, in the case of high rates of RNA survival in various RNA ages, the curves of RNA levels were smoother, such as RNA levels of HSP26 gene (Fig 1A) and GRX1 gene (S1A Fig). The RNA degradation rates were similar to the transcription rates due to small differences between cumulative transcription and degradation levels. The experiment duration covered a few RNA lifespans; hence, most of RNAs transcribed during the experiment were degraded by the end of the experiment.

## mRNA level fluctuations in a cell in response to various stimuli

The results of the simulation revealed dynamic cycles of the total level of RNA (TR) and total RNA activity level (TRA) under stable demands for RNA activity (DRA) (Fig 2), and these observations were consistent with the single-cell gene expression patterns reported in genetic laboratory studies shown in Figs 1B and 2A [11,21]. If TRA was less than DRA, nascent transcripts were produced, increasing TRA even though RNAs were simultaneously degraded. When TRA was higher than DRA, transcription stopped and RNA degradation continued; TRA decreased until it became lower than DRA. Age differences in RNA activity, RNA survival and RNA level at age 0 influenced the cycle lengths of TR and TRA oscillations. Parabolic changes in RNA activity coefficients caused TRA to respond slowly to DRA compared to the response corresponding to the highest RNA activity coefficient of 1, for example, by leading to an extension of the cycle length from 5 to 8 RNA ages (Fig 2B and 2C). TRA was decreased rapidly if RNA survival rates were lower, resulting in shorter lengths of fluctuating cycles (Fig 2D). Introduction of a maturation period for RNA activity, which was incorporated into parameter design by assigning lower activities of RNAs to early ages (Type C and Type D RNA activity coefficients in S2 Table), caused a delay in the TRA peak. Thus, TRA peaks occurred later than TR peaks of the oscillations (Fig 2C). Transcription rates were influenced by the frequencies of transcription pulses. During a cycle of RNA level fluctuation, the differences between RNA degradation levels and transcription levels were detected; however, all RNAs were degraded, resulting in equal levels of degradation and transcription at the end of the cycle. Therefore, at the end of the simulation for a few cycles of RNA levels, the RNA degradation rates were close to transcription rates (Fig 2B–2D). Different DRA values caused by differences in the levels of environmental stimuli were able to alter the amplitudes and cycling lengths of TR oscillations. The wavelength of TR with a DRA of 80 was equal to 7 RNA ages, and that with a DRA of 50 was equal to 5 RNA ages. The oscillation amplitude of TR with a DRA of 80 was equal to 96 and that with a DRA of 50 was equal to 85. The differences in the TR values with DRA values of 80 and 50 dramatically fluctuated, and TR with a DRA of 80 could be either higher or lower than that with a DRA of 50 (Fig 2E). Use of different sampling times may lead to incorrect results (Fig 2F). According to the simulation results, a sampling interval of 1 RNA age enabled to obtain correct oscillatory details, i.e., a wavelength of 8 RNA

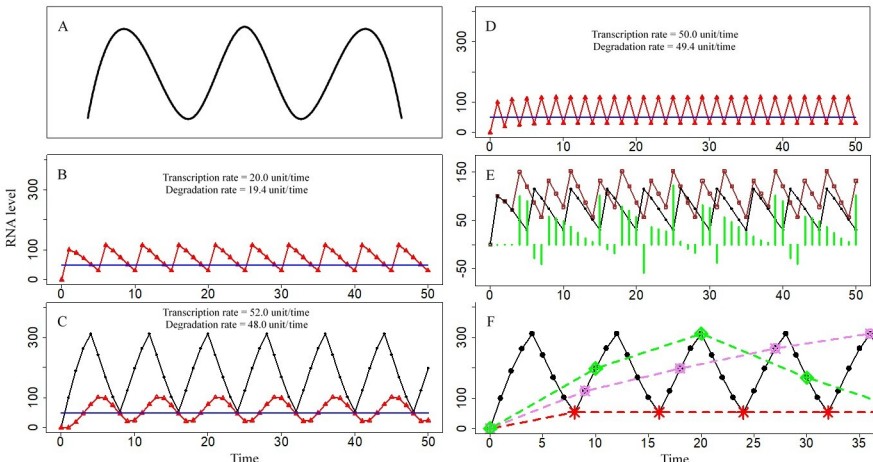

**Fig 2. Cycles in RNA level fluctuations.** Blue line represents the demand for RNA activity (DRA), which is set to 50. Black curve represents the total RNA amount (TR) with a DRA of 50, and points on the line are values predicted by the model. Brown curve represents TR with a DRA of 80, and hollow squares on the curve are predicted values. Red curve represents total RNA activity (TRA), and triangles on the curve are predicted values. Vertical green lines are the differences between TR with a DRA of 50 and TR with a DRA of 80. The x-axis is time, and the unit of time is RNA age, which is standardized as unitless values. The y-axis is the RNA level, which is standardized as unitless values. (A) Validation data: RNA level fluctuations reported in the literature [11,21], as shown in Fig 1B. (B) RNA level fluctuations with a cycle length of 5 RNA ages. TR and TRA curves overlap (DRA is set to 50. If TRA<DRA, then RNA level at age 0 = 100; otherwise, RNA level at age 0 = 0. The survival rate decreases at older RNA ages, i.e., the type B RNA survival rate is shown in S2 Table. The RNA activity coefficient of each RNA age is 1, i.e., the type A RNA activity coefficient in S2 Table). (C) RNA level fluctuations with cycle length longer than that in Fig 2B, resulting from reduced RNA activity coefficients (DRA is set to 50. If TRA<DRA, then RNA level at age 0 = 100; otherwise, RNA level at age 0 = 0. The survival rate decreased at older RNA ages, i.e., the type B RNA survival rate is shown in S2 Table. The RNA activity coefficient has a parabolic trend, i.e., the type C RNA activity coefficient in S2 Table). (D) RNA level fluctuations with cycle length shorter than that in Fig 2B, resulting from decreased RNA survival rates. TR and TRA curves overlap (DRA is set to 50. If TRA<DRA, then RNA level at age 0 = 100; otherwise, RNA level at age 0 = 0. The survival rate had a parabolic trend, i.e., the type C RNA survival rate is shown in S2 Table. The RNA activity coefficient of each RNA age is 1, i.e., the type A RNA activity coefficient in S2 Table). (E) Differences between TR with a DRA of 50 and TR with a DRA of 80. Green vertical lines are calculated differences between TRs with a DAR of 80 and TRs with a DRA of 50 (If TRA<DRA, then RNA level at age 0 = 100; otherwise, RNA level at age 0 = 0. The survival rate decreases at older RNA ages, i.e., the type B RNA survival rate is shown in S2 Table. The RNA activity coefficient of each RNA age is 1, i.e., the type A RNA activity coefficient in S2 Table). (F) Loss of oscillatory details in the results using various sampling times. Black solid line represents a sampling interval of 1 RNA age, and the points on the line are correct values. Red dashed line represents a sampling interval of 8 RNA ages. Violet dashed line represents a sampling interval of 9 RNA ages. Green dashed line represents a sampling interval of 10 RNA ages (DRA is set to 50. If TRA<DRA, then RNA level at age 0 = 100; otherwise, RNA level at age 0 = 0. The survival rate decreases at older RNA ages, i.e., the type B RNA survival rate is shown in S2 Table. The RNA activity coefficient of each RNA age is 1, i.e., the type A RNA activity coefficient is shown in S2 Table).

ages and a peak of 313 units; a sampling interval of 8 RNA ages resulted in a stable RNA level of 53 units; a sampling interval of 9 RNA ages resulted in a wavelength of 71 RNA ages; and a sampling interval of 10 RNA ages resulted in a wavelength of 39 RNA ages.

Environmental variations resulted in changes in DRA. Cyclic DRAs were able to induce new cycles of RNA dynamics (Fig 3). When DRA was changed from a stable value of 50 (Fig 2B) to a cyclic fluctuation between 50 for 5 RNA ages and 150 for 5 RNA ages, the cycle length of TR fluctuation changed from 5 RNA ages to 30 RNA ages (Fig 3B). The cycle lengths of TR and/or TRA were sometimes identical to those of cyclic DRA (Fig 3C). Aperiodic DRA eliminated the oscillations of RNA dynamics (Fig 3D). Simulated outcomes in which TR fluctuation was influenced by environmental variations matched the single-cell gene expression patterns reported in other studies, as shown in Figs 3A [12] and 1C [37] and S1C [21]. Sub-cycles of RNA dynamics were expected to occur during sufficiently long stable periods of cyclic DRA.

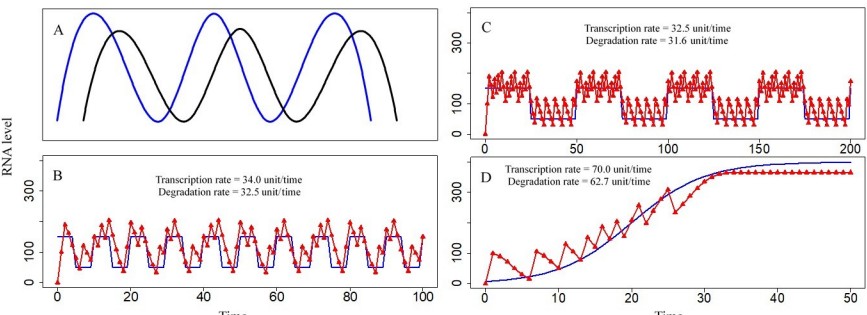

**Fig 3. Environmental variations influencing RNA level fluctuations.** Blue line represents the demand for RNA activity (DRA). Black curve represents total levels of RNA (TR). Red curve represents total RNA activity (TRA), and triangles on the curve are predicted values. The x-axis is time, and the unit of time is RNA age, which is standardized as unitless values. The y-axis is the RNA level, which is standardized as unitless values. (A) Validation data: RNA level fluctuations with oscillatory environmental levels reported in the literature [12] and RNA level fluctuations with irregular environmental effects shown in Figs 1C [37] and S1C [21]. (B) Differences in the cycle lengths of TR and DRA. TR and TRA curves overlap (DRA is rotated between 50 for 5 RNA ages and 150 for 5 RNA ages. If TRA<DRA, then RNA level at age 0 = 100; otherwise, RNA level at age 0 = 0. The survival rate decreases at older RNA ages, i.e., the type B RNA survival rate is shown in S2 Table. The RNA activity coefficient of each RNA age is 1, i.e., the type A RNA activity coefficient in S2 Table). (C) Similarity in the cycle lengths of TR and DRA, with local sub-cycles within the global cycles. TR and TRA curves overlap (DRA is rotated between 50 for 25 RNA ages and 150 for 25 RNA ages. If TRA<DRA, then RNA level at age 0 = 100; otherwise, RNA level at age 0 = 0. The survival rate decreases at older RNA ages, i.e., the type B RNA survival rate is shown in S2 Table. The RNA activity coefficient of each RNA age is 1, i.e., the type A RNA activity coefficient in S2 Table). (D) Aperiodic DRA. TR and TRA curves overlap (DRA is a logistic function: DRA = $400/(1+e^{4-0.2 \times time})$. If TRA<DRA, then RNA level at age 0 = 100; otherwise, RNA level at age 0 = 0. The survival rate decreases at older RNA ages, i.e., the type B RNA survival rate is shown in S2 Table. The RNA activity coefficient of each RNA age is 1, i.e., the type A RNA activity coefficient in S2 Table).

Three cycles in two hierarchies were observed if DRA alternated between 50 for 25 RNA ages and 150 for 25 RNA ages. A subcycle of 5 RNA ages was detected during stable periods with a DRA of 50, and a subcycle of 7 RNA ages was detected during stable periods with a DRA of 150; additionally, a global cycle of 50 RNA ages was detected over time (Fig 3C). Several responses to reach various DRAs were detected. For example, TRA corresponded to DRA fluctuations (Fig 3B and 3C) or to a reduced DRA without reaching a higher DRA (Fig 3D). If fluctuations in the attainable DRA maintained regular cycles, then TR and TRA were typically driven into regular cycling dynamics. Altered DRAs influenced the frequencies of transcriptional pulses, causing variable transcription rates. When TRAs were unable to reach DRAs, continuous transcriptional pulses resulted in higher transcription rates (Fig 3D) compared to low transcription rates in the case of attainable DRAs (Fig 3B and 3C).

In some cases, TR and TRA gradually changed to reach stable levels over time (Fig 4), and this observation was consistent with reported data shown in Figs 4A and S1B [11,21,38]. If transcription continued despite a DRA limit, i.e., transcription was unregulated, TR and TRA were adjusted via transcription and degradation to reach equilibrium (Fig 4B). When TRA was less than DRA, the TR and TRA values gradually stabilized over time (Fig 4C and 4D). On the one hand, if RNA activity coefficients were low, TRA was also low despite a high TR. Under these conditions, despite a low DRA, TRA was less than DRA, and straight lines corresponding to the TR and TRA values were subsequently achieved (Fig 4C). On the other hand, excessively high DRAs made it impossible for TRA to reach DRA (Fig 4D). Unregulated transcription or unattainable DRA resulted in continuous transcription and higher transcription rates (Fig 4B–4D) compared to those detected under other conditions (Figs 2 and 3). The

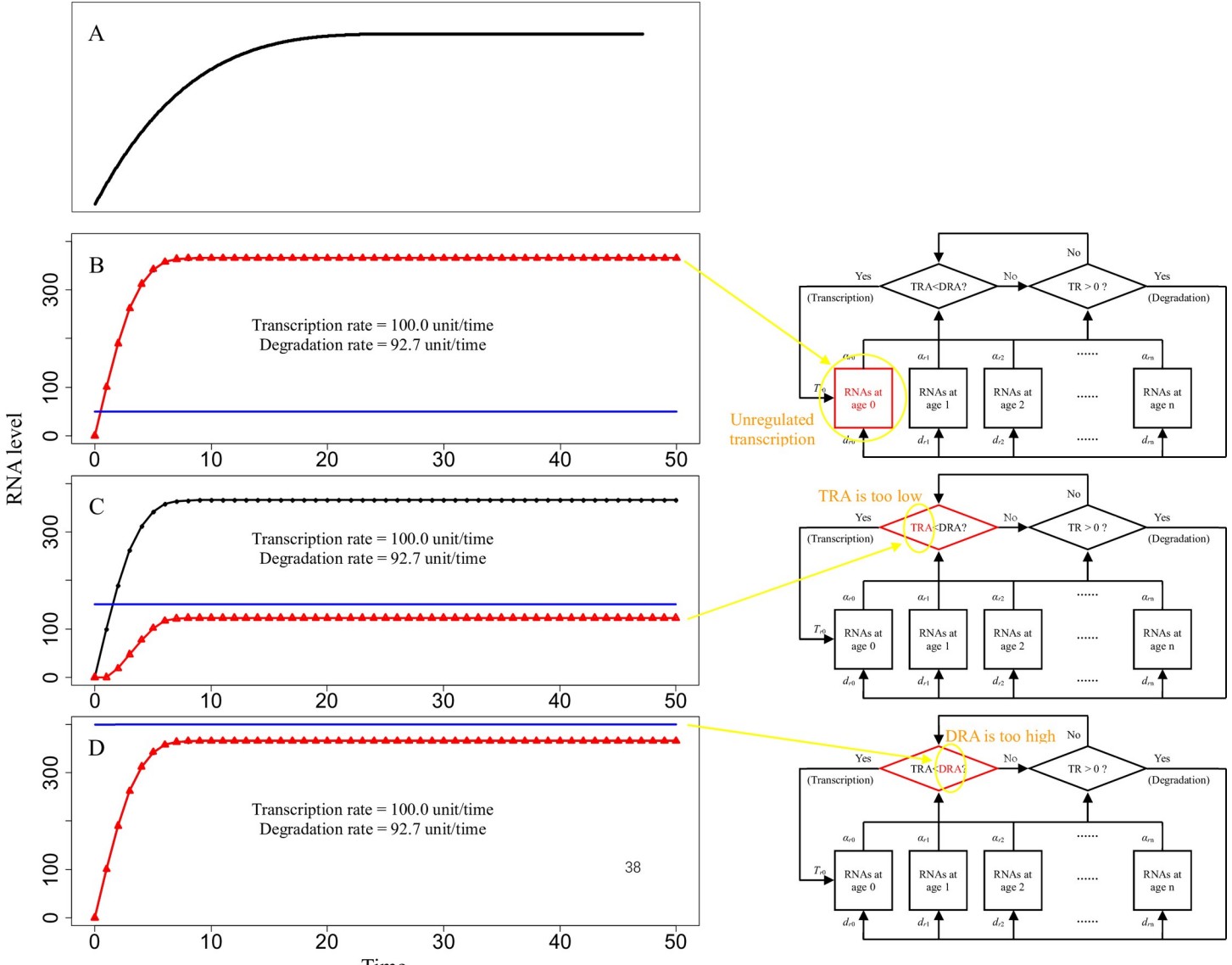

**Fig 4. Stable RNA levels.** Blue line represents the demand for RNA activity (DRA). Black curve represents the total amount of RNA (TR), and points on the line are predicted values. Red curve represents total RNA activity (TRA), and triangles on the curve are predicted values. $d_{r0}, d_{r1}, d_{r2}, \ldots, d_{rn}$ are degradation coefficients of RNA. $\alpha_{r0}, \alpha_{r1}, \alpha_{r2}, \ldots, \alpha_{rn}$ are the activity coefficients of RNA. $T_{r0}$ is the amount of nascent RNA per time. The x-axis is time, and the unit of time is RNA age, which is standardized as unitless values. The y-axis is the RNA level, which is standardized as unitless values. (A) Validation data: stable RNA level reported in the literature [11,21,38], as shown in S1B Fig. (B) Transcription is unregulated. TR and TRA curves overlap (DRA is 50. The level of the transcripts at age 0 is 100 per transcript during the whole experiment. The survival rate decreases at older RNA ages, i.e., the type B RNA survival rate is shown in S2 Table. The RNA activity coefficient of each RNA age is 1, i.e., the type A RNA activity coefficient in S2 Table). (C) TRA that is too low (DRA is 150. If TRA<DRA, then RNA level at age 0 = 100; otherwise, RNA level at age 0 = 0. The survival rate decreases at older RNA ages, i.e., the type B RNA survival rate is shown in S2 Table. The RNA activity coefficient has a parabolic trend, i.e., the type C RNA activity coefficients in S2 Table). (D) DRA that is too high. TR and TRA curves overlap (DRA is 400. If TRA<DRA, then RNA level at age 0 = 100; otherwise, RNA level at age 0 = 0. The survival rate decreases at older RNA ages, i.e., the type B RNA survival rate is shown in S2 Table. The RNA activity coefficient of each RNA age is 1, i.e., the type A RNA activity coefficient in S2 Table).

ability of TR to ultimately reach a certain value and stabilize over time may have biological and medical significance. For instance, if a specific RNA is maintained at a stable level, the cell may fail to meet the demands for RNA or expression of the corresponding gene becomes unregulated. These events can result in selection of a specific RNA as a candidate target in the studies of gene function, disease diagnosis, drug design, or other aspects.

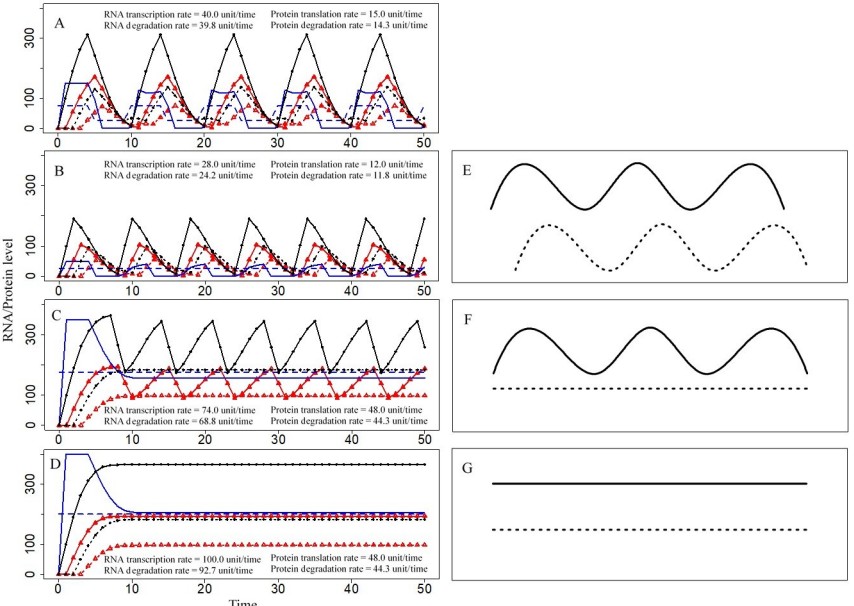

**Fig 5. Relationship between RNA and protein level fluctuations.** Blue curve represents the demand for RNA activity (DRA). Blue dashed line represents the demand for protein activity (DPA). Red curve represents total RNA activity (TRA), and triangles on the curve are predicted values. Red dashed line represents total protein activity (TPA), and triangles on the line are predicted values. Black curve with dots represents the total amount of RNA (TR), and points on the line are predicted values of RNA. Black dashed line represents the total amount of protein (TPro), and the points on the line are the predicted values of protein. The values of the parameters are given in S3 Table. The x-axis is time, and the unit of time is RNA/Protein age, which is standardized as unitless values. The y-axis is the RNA/protein level, which is standardized as unitless values. (A) Cycling TR and cycling TPro under the condition of cycling DPA (DPA rotated between 25 and 75). (B) Cycling TR and cycling TPro under the condition of stable DPA = 25. (C) Cycling TR and stable TPro under the condition of stable DPA = 175. (D) Stable TR and stable TPro under the condition of stable DPA = 200. (E) Cycling TR and cycling TPro in the literature [39]. (F) Cycling TR and stable TPro in the literature [39]. (G) Stable TR and stable TPro in the literature [39].

## The relationship between RNA and protein level fluctuations

In the two-unit model, the levels of protein and RNA were driven by the demand for protein activity (DPA) (Fig 5). When DPA fluctuated over time, TR and the total amount of protein (TPro) also followed similar trends (Fig 5A). When the DPA value was stable, TPro underwent fluctuating cycles, accompanied by fluctuating cycles in TR that resulted from DRA cycling (Fig 5B). If total protein activity (TPA) did not reach DPA over time, TPro changed to a constant value. However, two trends of the changes were possible. In one case, TRA exceeding DRA resulted in the changes in the dynamic cycles of TR (Fig 5C); if TRA was less than DRA over time, the TR value also stabilized (Fig 5D). The relationship between RNA and patterns of protein dynamics was consistent with that reported in other studies (Fig 5E–5G) [39].

In the simulation experiments, the protein translation rates peaked at 48 units/time if TPA did not reach DPA and Tpro was stable (Fig 5C and 5D), and the RNA transcription rates peaked at 100 units/time if TRA did not reach DRA and TR was stable (Fig 5D). In the case of simulation experiments with attainable DPA and fluctuating Tpro, the protein translation rates varied within a narrow range of 12~15 units/time (Fig 5A and 5B); however, in the case of attainable DRA and fluctuating TR, the RNA transcription rates varied within a wider range of 28~74 units/time (Fig 5A–5C). The trends of the RNA degradation rates were similar to those of the transcription rates, and the protein degradation rates demonstrated similar trends to those of protein translation rates.

The results of the simulations indicated that the values of the Tpro/TR ratios changed dramatically, from 0 to positive infinity; however, the ratios of protein translation rates to RNA transcription rates maintained relatively stable values of 0.38~0.65.

## Discussion

### Significance of mRNA fluctuations

Dynamic cycling of total biomolecule levels (TB) occurred either under the conditions of stable demand for biomolecule activities (DA) over time (Fig 2B) or was embedded in the stable phase of global fluctuations (Fig 3C). If a stable DA originates from a basic cellular requirement, intrinsic circular rhythms may appear. Circular changes in environmental stimuli form circular DAs, leading to some TBs with the same rhythms (Fig 5A); this result was consistent with reports on RNA rhythms induced by intrinsic and/or extrinsic elements, such as light, food, hormone signals, and cell-cell communication [12,19,20,40]. Therefore, intrinsic circular rhythms of important biomolecules act as environmental stimuli that influence other biomolecules to establish biological rhythms in the cells, tissues, or even whole organisms. This scenario may also explain some aspects of the mechanism underlying biological rhythms [22,41].

A positive feedback loop coupled with negative feedback factors partially explains the mechanism of circadian fluctuations in biomolecules [7,42]. Similarly, in our model, environmental stimuli, such as DRA, replaced the positive feedback mechanism to trigger transcription, and RNA degradation acted as a negative feedback factor. Stochastic pulsing in transcription has been reported in many studies [8,9,14,23,31–33,43–45]. Therefore, transcriptional pulsing was used in our model. Reports on stochastic transcription suggested that stochasticity results from certain unknown determinants, such as cell cycle variability [14,23] and stochastic DRAs, leading to stochastic RNA levels observed in our model.

### Estimation of RNA metabolic rates, RNA age distribution, and other parameters

RNA metabolic rates, such as net RNA transcription rates and net RNA degradation rates, contain important biological and medical information; however, investigations of these aspects are limited by technological methods [27,30,35,36] and are difficult to perform in humans and living organisms. RNA abundance data have been produced in very many studies in recent decades; however, direct use of RNA abundance as net transcription rates or net RNA production levels can easily result in incorrect evaluation [34]. Given sufficient time series RNA abundance data, the present model successfully estimated the transcription rates, RNA degradation rates, RNA demands, RNA life spans, accumulated transcription levels, accumulated RNA degradation levels, etc. (Fig 1). Certain parameters, e.g., the demand for RNA activity (DRA), of the model need reasonable explanation. In the case of the expression of the HSP26 gene (Fig 1A) and YNR014W gene (Fig 1B), DRAs were closely related to exogenous stimulatory signals of 1-NM-PP1 [21]. Detailed biological significance of a DRA is sometimes difficult to estimate due to limited information on the stimuli, such as analytic results on the GnRH gene in mouse GT1-1 cells (Fig 1C).

RNA age influenced RNA degradation rates, which determined the presence of TR cycling and the shapes of the response curves of RNA fluctuations. Age-dependent differences in RNA degradation determined the maxima of TR, which in turn influenced whether TRA reaches the DRA. If low RNA survival rates induced a rapid decrease in TR, the wavelengths of the fluctuation curves were lower (Fig 2D). Some information on the processes and effects of RNA aging, such as posttranscriptional processing of RNA and the identification of RNAs

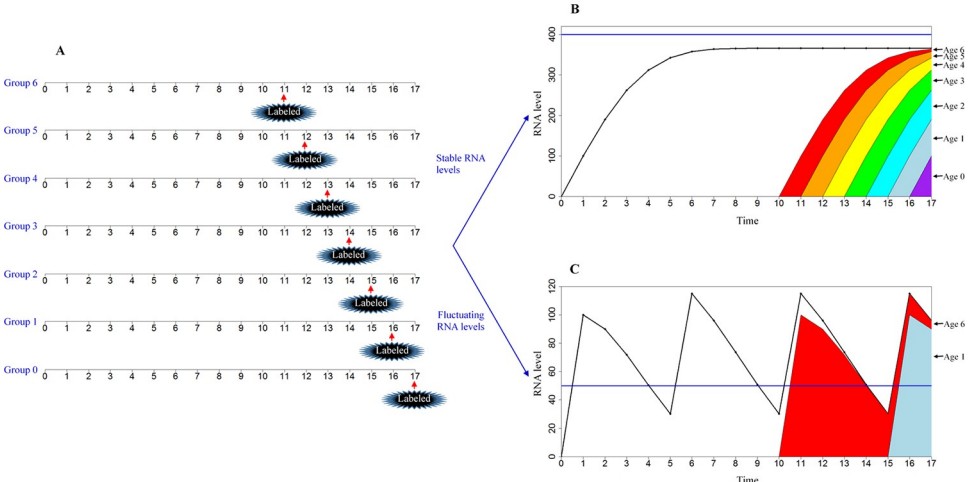

**Fig 6. Measurement of RNA age distribution based on the present model.** (A) Design of the experiment. Seven groups of the cells were used in the experiments. The cells were cultured in unlabelled normal media at time 0 and exposed to labelled media at time 11 in Group 6, at time 12 in Group 5, at time 13 in Group 4, at time 14 in Group 3, at time 15 in Group 2, at time 16 in Group 1, and at time 17 in Group 0. The samples were collected from time 0 to time 17, and RNA levels (of one gene or some genes) were detected. At the time of exposure in labelled media and later, unlabelled RNAs and labelled RNAs were separated and detected. The RNA level at a certain RNA age at a certain time point was calculated based on the differences between labelled RNA levels in two adjacent groups. Consider RNA levels at time 17 an example. RNA level at age 6 = labelled RNA level in Group 6 –labelled RNA level in Group 5, RNA level at age 5 = labelled RNA level in Group 5 –labelled RNA level in Group 4, RNA level at age 4 = labelled RNA level in Group 4 –labelled RNA level in Group 3, RNA level at age 3 = labelled RNA level in Group 3 –labelled RNA level in Group 2, RNA level at age 2 = labelled RNA level in Group 2 –labelled RNA level in Group 1, RNA level at age 1 = labelled RNA level in Group 1 –labelled RNA level in Group 0, and RNA level of age 0 is labelled RNA level in Group 0. (B) RNA age distribution under stable RNA levels. Blue line represents the demand for RNA activity (DRA). Black curve represents the total amount of RNA (TR). Colours indicate levels of RNAs labelled at various time points, and levels of RNAs labelled at later times cover the levels of RNA labelled at earlier times. The RNA levels at age 6, age 5, age 4, age 3, age 2, Age 1, and age 0 were 6 (= 364–358), 15 (= 358–343), 31 (= 343–312), 50 (= 312–262), 72 (262–190), 90 (190–100), and 100, respectively (DRA is 400. If total RNA activity (TRA)<DRA, then RNA level at age 0 = 100; otherwise, RNA level at age 0 = 0. The survival rate decreases at older RNA ages, i.e., the type B RNA survival rate is shown in S2 Table. The RNA activity coefficient of each RNA age is 1, i.e., the type A RNA activity coefficient in S2 Table). The x-axis is time, and the unit of time is RNA age, which is standardized as unitless values. The y-axis is the RNA level, which is standardized as unitless values. (C) RNA age distribution under conditions of fluctuating RNA levels. The blue line represents DRA. Black curve represents TR. Colours indicate the levels of RNAs labelled at various time points, and levels of RNA labelled at later times cover the levels of RNA labelled at earlier times. The RNA levels at age 6, age 5, age 4, age 3, age 2, age 1, and age 0 were 6 (= 96–90), 0 (= 90–90), 0 (= 90–90), 0 (= 90–90), 0 (= 90–90), 90 (90–0), and 0, respectively (DRA is set to 50. If TRA<DRA, then RNA level at age 0 = 100; otherwise, RNA level at age 0 = 0. The survival rate decreases at older RNA ages, i.e., the type B RNA survival rate is shown in S2 Table. The RNA activity coefficient of each RNA age is 1, i.e., the type A RNA activity coefficient in S2 Table). The x-axis is time, and the unit of time is RNA age, which is standardized as unitless values. The y-axis is the RNA level, which is standardized as unitless values.

transcribed in series transcription [29,36], is available; however, further investigation of finer details is needed.

How to measure RNA age is a new question. An RNA timestamp approach was previously used to infer the age of individual RNAs transcribed via the same promoter [26]; however, this method cannot directly measure the levels of RNAs with various ages at the same time. We think that a labelling method can solve this problem. RNAs can be labelled, and the labelled RNAs can be isolated and sequenced [35,36]. Suppose that several treatment groups are used in a simulation experiment, and the cells of each group are exposed to the labelled media at various time points based on the present model (Fig 6). The levels of RNA at a certain RNA age can be calculated based on the differences between labelled RNA levels in various groups. The results of an experiment measuring RNA age can be predicted by the present model. In

the case of stable RNA levels, cells will have RNAs with consecutive ages, and the levels of RNAs at each age are in a dynamic equilibrium state (Fig 6B). In the case of fluctuating RNA levels, the cells will have RNAs with inconsecutive ages because RNAs are not produced when TRA is greater than DRA; the levels of RNAs at each age fluctuate (Fig 6C). Additional relevant experiments are expected to measure RNA age distributions.

## Effects of RNA fluctuations on the analysis of transcriptomic data

Fluctuations in cellular RNA abundance can introduce certain challenges for selection of sampling time points and comparisons between the treatment and control groups. If RNA levels were stable (Fig 4), sequential sampling, or even one-time sampling, may yield reasonable results. However, when RNA levels oscillate, the measurement of time samples at a single time point, and even sequential sampling, may yield random results. At the early stages of an experiment, the amount of RNA in the cells before the start of the experiment may substantially influence the results (Figs 2–4); after this point, the levels of biomolecules were regular. If the levels were oscillatory at the regular stages, the results may depend on the design of time intervals for sequential sampling (Fig 2F). If the sampling interval equals the wavelength of the cycling curve of a biomolecule, the results of the experiment appeared to be a level line on a curve. Perfect and thorough sampling should cover all feature points of an oscillatory curve. Comparison of the treatment groups with the controls is the gold standard for biological experiments when measured values are nearly constant (Fig 4), and statistical comparison between the groups is straightforward. However, in some cases of cyclical abundance of a biomolecule, the levels of biomolecules experiencing high environmental pressures (e.g., the results of the treatment groups) may be either higher or lower than those experiencing low environmental pressures (e.g., the results of the control groups) depending on the time of sampling (Fig 2E). Indeed, in the present study, environmental pressures influenced the length of the cycle and vibration amplitudes of dynamics of biomolecule abundance (Fig 2E).

It was necessary to distinguish between transcriptional pulses and pulse-like fluctuations in RNA abundance. In our model, transcription occurred in pulses. After a transcriptional pulse, RNA levels abruptly increased; at this point, in the case of subsequent rapid degradation, RNA abundance dramatically decreased, and a pulse RNA level was detected (Fig 2B and 2D). If TRA did not reach DRA, pulse transcription continued, and RNA abundance successively increased; if TRA was higher than DRA, transcription stopped, and degradation continued, forming a pulse-like curve of RNA abundance (Fig 2C). Without degradation, a single pulse resulted in a horizontal line in the RNA abundance curve, and successive pulses resulted in an ascending curve (cumulative RNA transcription levels in Figs 1 and S1). Thus, transcriptional pulses do not always result in pulse or pulse-like curves of RNA abundance.

## The relationship between RNA and protein level fluctuations

If details of the parameters used in the present model are known, the protein levels can be calculated based on RNA levels (to some extent), and vice versa. Unfortunately, collecting enough information on numerous required parameters is not easy, and most of the parameters have not been studied in detail. The protein/mRNA ratios or translation rates of specific genes were previously shown to be constant, and protein levels were predicted based on mRNA levels [46,47]; however, these conclusions have been questioned [39,48]. In the present study, the protein/RNA ratios were likely to decrease within narrow ranges in some cases or even remained constant, especially under the conditions when TRA and TPA did not meet the requirements (Fig 5D). However, cycling levels of RNAs and proteins formed a range of the protein/RNA ratios from zero to infinity. The protein levels were stable; however, the protein/

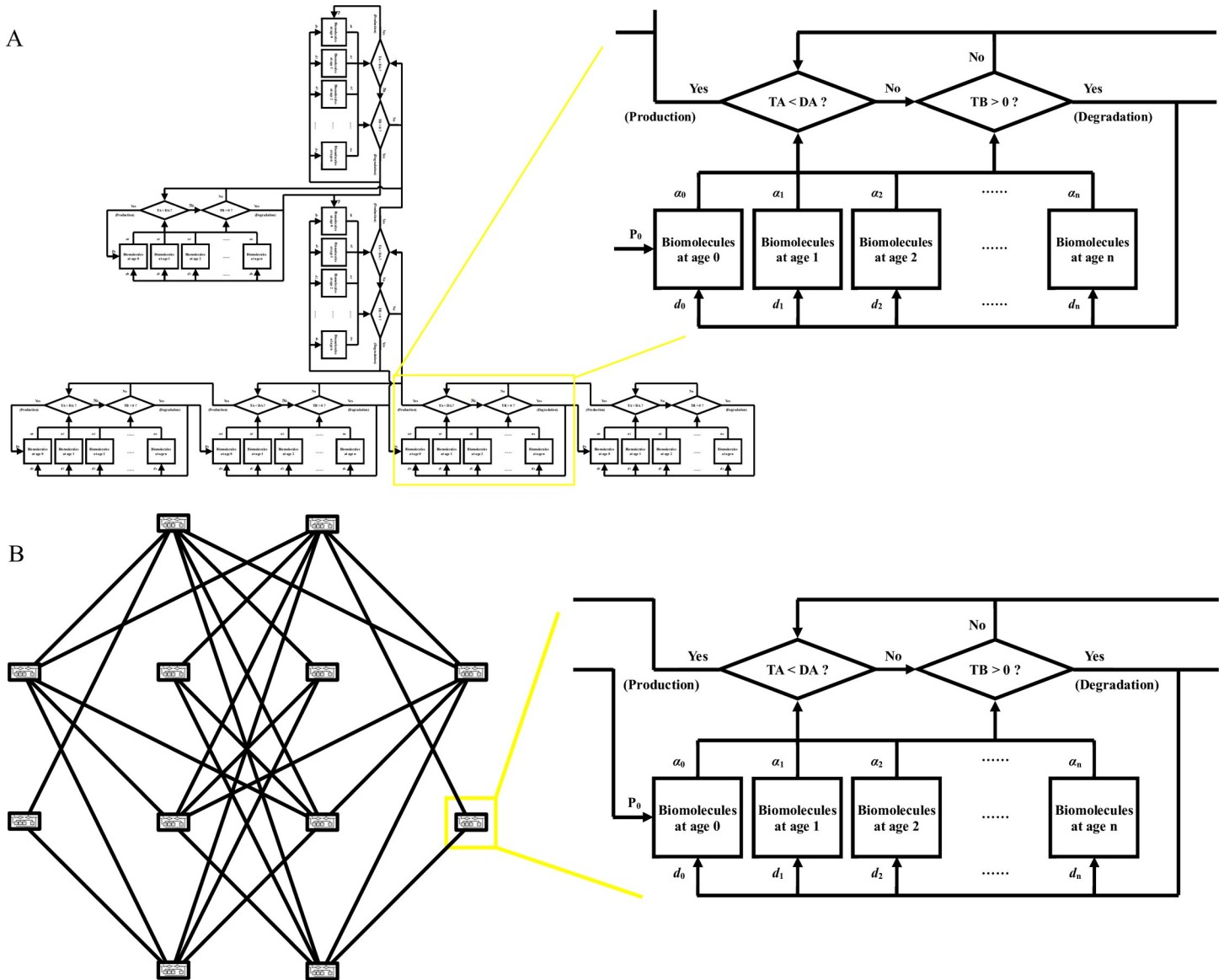

**Fig 7. Chain and web of the present model.** TB represents the total abundance of biomolecules at all biomolecular ages. TA represents total biomolecular activity at all biomolecular ages. DA represents the demands for biomolecular activity in the cell. DA is influenced by stimuli from intra- and extracellular environments, and the difference between TA and DA determines the trigger or cessation of biomolecule production. $d_0, d_1, d_2, \ldots, d_n$ are degradation coefficients. $\alpha_0, \alpha_1, \alpha_2, \ldots, \alpha_n$ are activity coefficients. $P_0$ represents the level of nascent biomolecules per time. (A) Chain of the present model. (B) Web of the present model.

RNA ratios were likely to vary due to fluctuating RNA levels (Fig 5C). Thus, the application of the protein/RNA ratios requires consideration of the relationships between fluctuations in RNA and protein levels.

RNA and protein abundance at a certain time point are snapshots, lacking information on the rates and cumulative amounts of transcription and translation during the experiment; hence, the ratios of these snapshots are difficult to predict and apply in practice. In contrast, the RNA transcription and protein translation rates contain information on biological processes. Thus, the ratios of protein translation rates to RNA transcription rates are more important for the analysis of research data on integration of the transcriptome and proteome. Moreover, in our model, the ratios of the protein translation rates to RNA transcription rates

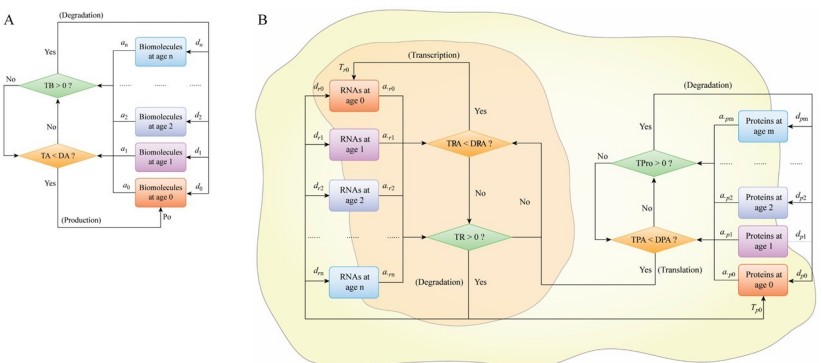

**Fig 8. Models of biomolecular level fluctuations.** (A) A one-unit model of biomolecular level fluctuations in a cell. TB represents the total abundance of biomolecules at all biomolecular ages. TA represents total biomolecular activity at all biomolecular ages. DA represents the demands of biomolecular activity in the cell. DA is influenced by stimuli from intra- and extracellular environments, and the difference between TA and DA determines the trigger or cessation of biomolecule production. $d_0, d_1, d_2, \ldots, d_n$ are degradation coefficients. $\alpha_0, \alpha_1, \alpha_2, \ldots, \alpha_n$ are activity coefficients. $P_0$ represents the level of nascent biomolecules per time. (B) A two-unit model of level fluctuations of RNAs and proteins derived from a gene in a cell. TR is the total amount of RNA at all ages. TRA is the total RNA activity at all ages. DRA is the demands for RNA activity. TPro is the total amount of protein at all ages. TPA is the total protein activity at all ages. DPA is the demands for protein activity. DPA corresponds to the external and internal stimuli. The differences between TPA and DPA and that between TRA and DRA define the intensity of the production and/or the degradation of proteins and RNAs, respectively. $d_{r0}, d_{r1}, d_{r2}, \ldots, d_{rn}$ are degradation coefficients of RNA. $d_{p0}, d_{p1}, d_{r2}, \ldots, d_{pn}$ are degradation coefficients of protein. $\alpha_{r0}, \alpha_{r1}, \alpha_{r2}, \ldots, \alpha_{rn}$ are the activity coefficients of RNA. $\alpha_{p0}, \alpha_{p1}, \alpha_{p2}, \ldots, \alpha_{pn}$ are the activity coefficients of the protein. $T_{r0}$ is the amount of nascent RNA per time. $T_{p0}$ is the amount of nascent protein per time.

varied within a narrower range (0.38~0.65) compared to the enormous range of the Tpro/TR ratio values (0~+∞). These relatively stable ratio values may be extensively used to calculate the protein translation rates or protein amounts based on the RNA transcription rates. Although the specific values of the ratio of the protein translation rates to RNA transcription rates in the present study were closely related to designed simulation parameters (S3 Table), we believe that future studies may be able to consolidate, revise, and/or expand the list of the ratio values for expanded and general applications.

## Conclusions

Introduction of biomolecule age and the demands for biomolecule activity to our model enabled to explain the mechanism underlying fluctuations in cellular RNA levels detected in previous studies based on the cooperation between the production and degradation. Important transcriptomic parameters, such as the transcription rates and degradation rates, of time-series data on gene expression data can be evaluated using this model.

Some new hypotheses were generated based on the analytical results under various simulation environments. Demands for biomolecule activity govern the frequencies of biomolecule production, resulting in an increase in the biomolecule production rates. Instead of the Tpro/TR ratios, the ratios of the protein translation rates to RNA transcription rates may be appropriately used to estimate the protein levels based on RNA levels during a certain period. Various treatment levels corresponding to the biomolecule ages and activities can change the wavelength and oscillation amplitude of oscillating RNA or protein levels. Thus, selection of the sampling time points and the results obtained based on typical assumptions for the parameters of genetic up- and downregulation in transcriptomic experiments must be carefully considered. Stable abundance of a specific RNA in a cell likely indicates that total activities of this RNA cannot meet the cellular requirements or that the production of this RNA is unregulated.

This information contributes to existing knowledge. Therefore, biological investigations into biomolecule age and biomolecule activity demands, and detailed value assignment for specific models require further research.

Multiunit models are additional applications of the present model (Fig 7). The levels and total activities of a specific biomolecule are influenced by other biomolecules and in turn regulate other biomolecules, forming a biomolecular relationship chain or biomolecular network with additional details. Incorporation of biomolecule ages and the demand for biomolecule activities into a study of metabolism of multiple biomolecules will assist in investigations of the mechanism of dynamic changes at the biomolecule level. This subject requires further study, especially because it involves a huge amount of computational work.

## Methods

### Description of the model

This model was built based on the production of biomolecules, including RNAs and proteins, triggered by the demand for biomolecule activities (DA) and their age-dependent degradation kinetics (Fig 8A). Certain studies reported the definitions of RNA age and half-life for RNA and protein [25,26,30], and numerous studies in molecular biology investigated the degradation and turnover of biomolecules, which are closely related to the age of these biomolecules. Thus, in the present study, to better describe possible degradation and biological activity of a biomolecule at various time points after its production, the age of a biomolecule was defined as the length of time this biomolecule existed from its production. The production of the biomolecules was induced by the stimuli. There was a maximum number of biomolecules at age 0 due to limits imposed by molecular effects and microspaces [12,24,39]. Subsequently, the biomolecules matured, aged, and were gradually degraded. The percentage of the survival rates was used to describe the degradation of the biomolecules. The age of biomolecule was considered to influence their survival and activity. We supposed that DA is a physiological reaction to a stimulus derived from the intrinsic or extrinsic environment and that the DA value reflects the stimulus level. In the model, if total existing activity of a biomolecule (TA) was less than DA, the production of biomolecules was triggered; otherwise, the production of biomolecules ceased. The abundance and activity of a biomolecule in this model were calculated as follows:

$$B_{0,t} = B0$$

$$B_{x,t} = B_{x-1,t-1} \times d_{x-1}$$

$$A_{x,t} = B_{x,t} \times a_x$$

$$TB_t = \sum_{x=0}^{n} B_{x,t}$$

$$TA_t = \sum_{x=0}^{n} A_{x,t}$$

where $B$ is the abundance of a biomolecule; $t$ is time; $B0$ is the abundance of biomolecules at age 0; $x$ is the age of the biomolecules; $d$ is the degradation coefficient (survival percentage relative to biomolecule abundance); $A$ is the activity of the biomolecules; $a$ is the activity coefficient; $TB$ is the total amount of biomolecules of all ages; $n$ is the maximum age of the biomolecules; and $TA$ is the total activity of the biomolecules of all ages.

## Simulation of single-cell experimental data and calculation of the transcription rates, RNA degradation rates, and other parameters

This model was used to simulate experimental data of mRNA level fluctuations (including regular, partially regular, and irregular fluctuations) in single cells (S1 Table) reported in the literature [21,37] by calculating the parameters, such as the transcription rates, RNA degradation rates, RNA demands, RNA life spans, RNA survival rated based on RNA ages, accumulated transcription levels, and accumulated RNA degradation levels. The exhaustive search method was used in the simulation, and the minimum square was the selection standard. To reduce the amounts of calculations with the exhaustive search method, the activity coefficients were set to 1; therefore, the RNA activity was equal to the RNA level, and the RNA activity demand was equal to the RNA demand. Five RNA ages, which divided an RNA span into four periods on average, were set. According to previous studies of transcriptional pulsing [8,32,33], we have also set transcriptional pulsing. The RNA lifespan, RNA demands, RNA survival rates at each RNA age, transcriptional pulsing level at the time when RNA abundance was lower than RNA demands, and RNA age distribution of RNA abundance at the beginning of the experiment were set as unknown variables to be assessed. To enable calculation using the exhaustive search method, two scenarios were set as follows: if RNA levels were stable or periodically fluctuated, stable RNA demands were set over time; if the single-cell experimental RNA levels randomly fluctuated, we used every 4 (or more than 4 if the computation ability of the computer permitted the settings) experimental expression values to assess the values of these variables and retain only the values of RNA demand. Finally, the values of RNA demand were set as known and other variables were assessed using all experimental expression values. In the exhaustive search method, the lower limit of the lifespan of RNA in yeast cells was set to 10 minutes, and the lower limit of the lifespan of RNA in mammalian cells was set to 2 hours according to the data of the literature [30].

The coefficient of determination ($R^2$) and the median absolute percentage error (MdAPE) were used to determine how well the results of the model simulation fit the experimental data. The equation for $R^2$ is:

$$R^2 = 1 - \frac{Unexplained\ variation}{Total\ variation}$$

MdAPE is the median of absolute percentage error, and the equation for absolute percentage error (APE) is:

$$APE = \left| \frac{Actual\ value\ of\ experiment\ data - Prediction\ value\ of\ the\ model}{Actual\ value\ of\ experiment\ data} \right|$$

## mRNA level fluctuations in single cells under various stimuli

A one-unit model was used to explain RNA fluctuations in a cell. The values of the following parameters were set to simulate the transcriptional features of a gene: the demand for RNA activity (DRA), RNA level at age 0, RNA survival rates, and RNA activity coefficients (S2 Table). Intra- and extracellular stimuli were incorporated into the DRA. Stable and fluctuating DRA values were both incorporated into the design. RNA lifespan was standardized and divided into 11 ages, from age 0 to age 10, when all RNAs were completely degraded. RNA ages influenced RNA survival and activity. The four types of relationships between RNA age and RNA survival or RNA activity were as follows: Type A, the parameter values for RNA survival or RNA activity were stable at all RNA ages; Type B, the parameter values decreased with increasing RNA age; Type C, the trend of the parameter values was parabolic; and Type D, the parameter values increased with increasing RNA age. Transcription was considered pulse-like

in nature. To enable comparison of the results of the simulation with the data from other sources, RNA levels and RNA activities were standardized as unitless values. Total RNA levels and total RNA activity levels (TRA) were the outputs of the model simulation.

General patterns of cellular RNA fluctuations have been described in a number of single-cell gene expression studies [4,5,6,20,21,37]. We used the data provided by these studies to verify the outputs of the model simulation.

## The relationship between mRNA and protein level fluctuations under various stimuli

A derivative two-unit model was built to explain the relationship between RNA and protein dynamics (Fig 8B). Extra-cellular and intra-cellular stimuli were incorporated into the demand for protein activity (DPA). If the total protein activity (TPA) generated by the total amount of protein (TPro) was less than DPA, the difference between the values was set as a stimulus to produce DRA, thereby driving changes in RNA. A set of parameter values in this two-unit model (S3 Table) was used to evaluate dynamic relationships between transcription and translation. The total amount of RNA (TR) and TPro were the outputs of the model simulation.

The relationship between cellular protein levels and mRNA abundance has been summarized in the data reported in the literature [39]. The relationships between mRNA and protein abundance under various conditions is complex, and mRNA abundance is not sufficient to predict protein abundance in many scenarios. After excluding the influence of protein translation delay, three categories of representative relationships between RNA and protein levels were considered: cycling RNA levels and cycling protein levels, cycling RNA levels and stable protein levels, and stable RNA levels and stable protein levels. We used the data provided by these studies to verify the outputs of the model simulation.

## Programming

The simulation processes were translated into C++ programs (S1 Code), and the simulation outputs (provided if requested) were recorded. Plotting functions of R program (S2 Code) were used to generate the curves of the simulation outputs.

## Supporting information

**S1 Fig. Three additional examples analysis of experimental data of single-cell RNA abundance using the present model.** Hollow circles represent RNA levels of experimental data obtained from the literature [21]. Blue line represents the demand for RNA activity (DRA). Red curve represents the RNA level based on the results of the model simulation. Black solid line represents the cumulative transcription RNA level calculated by the model. Black dashed line represents the accumulated degradation RNA level calculated by the model. The x-axis is time, and the unit of time is minute. The y-axis is the RNA level, and the unit is normalized fold change of mRNA level with the baseline subtracted. (A) Partially regular fluctuations: analysis of mRNA level fluctuations of the *Saccharomyces cerevisiae* GRX1 gene by the model. The estimated results of the parameters: accumulated transcription level = 6 units, accumulated RNA degradation level = 4.7 units, life span = 30 minutes, pulse transcription level at each time = 1 unit, the rate of survival at RNA age 1 = 0.6, the survival rate at RNA age 2 = 0.8, and the survival rate at RNA age 3 = 0.8. (B) Regular fluctuations: analysis of mRNA level fluctuations of the *S. cerevisiae* NTH1 gene by the model. The estimated results of the parameters: accumulated transcription level = 22 units, accumulated RNA degradation level = 22 units, life span = 10 minutes, pulse transcription level at each time = 1 unit, the rate of survival at RNA age 1 = 0.4, the survival rate of RNA age 2 = 0.8, and the survival rate of RNA age 3 = 0.2. (C)

Irregular fluctuations: analysis of mRNA level fluctuations of the *S. cerevisiae* CYB2 gene by the model. The estimated results of the parameters: accumulated transcription level = 11.9 units, accumulated RNA degradation level = 11.9 units, life span = 11 minutes, pulse transcription level at each time = 1.7 units, the rate of survival at RNA age 1 = 0.7, the survival rate of RNA age 2 = 0.9, and the survival rate of RNA age 3 = 0.1.
(TIF)

**S1 Table. Experiment data of single cell RNA abundance.**
(PDF)

**S2 Table. A set of parameter values used in the model to simulate RNA level fluctuations.**
(PDF)

**S3 Table. Parameter values in the model used to simulate the relationship between RNA and protein level fluctuations.**
(PDF)

**S1 Code. C++ codes.**
(TXT)

**S2 Code. R scripts for plotting.**
(TXT)

## Acknowledgments

We thank Dr. Ken Chan for his critical review of our work.

## Author Contributions

**Conceptualization:** Zhongneng Xu.

**Data curation:** Zhongneng Xu.

**Formal analysis:** Zhongneng Xu.

**Investigation:** Zhongneng Xu.

**Methodology:** Zhongneng Xu, Shuichi Asakawa.

**Project administration:** Zhongneng Xu.

**Resources:** Zhongneng Xu.

**Software:** Zhongneng Xu.

**Validation:** Zhongneng Xu.

**Visualization:** Zhongneng Xu.

**Writing – original draft:** Zhongneng Xu.

**Writing – review & editing:** Zhongneng Xu, Shuichi Asakawa.

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
