## [Decision Letter · Decision Letter 0]

5 Apr 2021

Dear Dr. XU,

Thank you very much for submitting your manuscript "A model explaining mRNA level fluctuations based on activity demands and RNA age" for consideration at PLOS Computational Biology.

As with all papers reviewed by the journal, your manuscript was reviewed by members of the editorial board and by several independent reviewers. In light of the reviews (below this email), we would like to invite the resubmission of a significantly-revised version that takes into account the reviewers' comments.

Please revise the manuscript carefully.

We cannot make any decision about publication until we have seen the revised manuscript and your response to the reviewers' comments. Your revised manuscript is also likely to be sent to reviewers for further evaluation.

Sincerely,

Quan Zou

Guest Editor

PLOS Computational Biology

Ilya Ioshikhes

Deputy Editor

PLOS Computational Biology

Please revise the manuscript carefully.

Reviewer's Responses to Questions

**Comments to the Authors:**

Reviewer #1: The authors developed a simulation model based on activity demands and RNA age to explore the mechanisms of RNA level fluctuations. The model used single-cell time series gene expression experimental data to determine the model parameters. The model was also able to predict RNA levels under simulation backgrounds, such as oscillation in RNA abundance, stable RNA levels and the relationship between RNA and protein levels and metabolic rates. The manuscript may benefit from the following revisions.

1. The axes of the figures need to be labeled.

2. How are the parameters for the two-unit model that’s used to explain the relationship between RNA and protein dynamics determined? Is there any difference with the exhaustive search method used for the RNA level model?

3. Figure 7 needs more elaboration in the main text.

Reviewer #2: In this study, the authors proposed a model to explain the mechanism underlying intracellular RNA level fluctuations. The overall framework of the model seems to be logical and up to the task. I would like to raise some general points regarding this manuscript:

1. Are there any references to support the definition of the age of a biomolecule used in this study?

2. Page 18, the formula of degradation coefficient s should be provided

3. R scripts for plotting should also provide

4. The Font in Figure 7 is too small to read. The authors are suggested to update this Figure

5. A brief description of the relationship between cellular protein levels and mRNA abundance is necessary, although it is summarised in the reported data (Liu et al., 2016)

Reviewer #3: In this study, Xu et al. proposed a computational model with pre-defined parameters to explore the mechanisms of RNA fluctuation in living cells. The authors showed that the model fits well to the real experimental data and can explain RNA level fluctuations in response to different stimuli. Generally, the study provides some novel insights and looks interesting to me. However, I have the following concerns need to be addressed:

- As my understanding, the authors fitted the parameters of their model using single-cell experimental data and showed that the parameters can predict values of the real experimental values well: “ R2 of each gene was over 0.9, indicating that the model predicted a good fit with these experimental values (line103-104)”. First, what’s the R2 here mean? The authors should make it clear to the reader how they evaluate the consistency. Second, it is not clear how they fit the parameters, usually one should fit the parameters in some data and see how it predict previous unseen data to show the performance. Here, I guess the parameters for different genes are different, so it is practical to leave some data points of certain gene out for cross-validation? If they just fit the parameters using all the data points and then predict the values, it’s not enough to make the statement “the model predicted a good fit with these experimental values”.

- The authors show three examples representing regular, partially regular and irregular fluctuations, which is good. However, to show the robustness of a computational model, it needs to be tested on data set at large scale. Considering that it may be difficult to perform a systematic evaluation in the current study, the author can at least provide one more example for each category in addition to the current one example. I found the data used in the study are from very old publications, is there any newer single-cell data available in larger scale?

- In line 490-493, the authors mentioned that “We used the data provided by these studies to verify the outputs of model simulations”. Yet, no details are provided about how they verify the outputs using the experimental data.

- Some figures, for examples, figure 2a, 3a, 4a, are hard to understand according to the figure legends, is it an example or just a schematic plot? The meaning of x-axis and y-axis is unclear. Further explanations about these figures in the legend are required. Fonts in figure 7a are too small.

- The writings of the manuscript seem unbalanced, I found the “Discussion” part interesting to read while the “Results” part very descriptive and boring. I would suggest that the authors put some of their own interpretations after describing the data in the “Results” instead of putting them in the “Discussion”.

**Have all data underlying the figures and results presented in the manuscript been provided?**

Reviewer #1: Yes

Reviewer #2: Yes

PLOS authors have the option to publish the peer review history of their article (what does this mean?). If published, this will include your full peer review and any attached files.

Reviewer #1: No

Reviewer #2: No

Reviewer #3: No

**Have the authors made all data and (if applicable) computational code underlying the findings in their manuscript fully available?**

Reviewer #3: Yes
---

## [Decision Letter · Decision Letter 1]

17 Jun 2021

Dear Dr. XU,

We are pleased to inform you that your manuscript 'A model explaining mRNA level fluctuations based on activity demands and RNA age' has been provisionally accepted for publication in PLOS Computational Biology.

Best regards,

Quan Zou

Guest Editor

PLOS Computational Biology

Ilya Ioshikhes

Deputy Editor

PLOS Computational Biology

Reviewer's Responses to Questions

**Comments to the Authors:**

Reviewer #1: By this revision, the authors have addressed the concerns raised in the review.

Reviewer #2: The authors addressed my comments

Reviewer #3: The authors have addressed all my concerns, and I think the manuscript is acceptable for publication.

**Have the authors made all data and (if applicable) computational code underlying the findings in their manuscript fully available?**

Reviewer #1: None

Reviewer #2: Yes

Reviewer #3: Yes

PLOS authors have the option to publish the peer review history of their article (what does this mean?). If published, this will include your full peer review and any attached files.

Reviewer #1: No

Reviewer #2: **Yes: **Fuyi Li

Reviewer #3: No

---

## [Editor Report · Acceptance letter]

9 Jul 2021

PCOMPBIOL-D-21-00340R1 

A model explaining mRNA level fluctuations based on activity demands and RNA age

Dear Dr XU,

I am pleased to inform you that your manuscript has been formally accepted for publication in PLOS Computational Biology. Your manuscript is now with our production department and you will be notified of the publication date in due course.

With kind regards,

Katalin Szabo
